# Partitioning surface ligands on nanocrystals for maximal solubility

Zhenfeng Pang [1,2], Jun Zhang[1,2], Weicheng Cao[1], Xueqian Kong [1] & Xiaogang Peng [1]

A typical colloidal nanoparticle can be viewed as a nanocrystal-ligands complex with an inorganic single-crystalline core, the nanocrystal, bonded with a monolayer of organic ligands. The surface chemistry of nanocrystal-ligands complexes is crucial to their bulk properties. However, deciphering the molecular pictures of the nonperiodic and dynamic organic-inorganic interlayer is a grand technical challenge, and this hampers the quantitative perception of their macroscopic phenomena. Here we show that the atomic arrangement on nanocrystal surface and ligand-ligand interactions can be precisely quantified through comprehensive solid-state nuclear magnetic resonance (SSNMR) methodologies. The analyses reveal that the mixed ligands of n-alkanoates on a CdSe nanocrystal segregate in areal partitions and the unique arrangement unlocks their rotational freedom. The mathematical model based on the NMR-derived ligand partition and dynamics successfully predicts the unusual solubility of nanocrystal-ligands complexes with mixed ligands, which is several orders of magnitude higher than that of nanocrystal-ligands complexes with pure ligands.

[1] Center for Chemistry of High-Performance & Novel Materials, Department of Chemistry, Zhejiang University, Hangzhou 310027, China. [2]These authors contributed equally: Zhenfeng Pang, Jun Zhang. Correspondence and requests for materials should be addressed to X.K. (email: kxq@zju.edu.cn) or to X.P. (email: xpeng@zju.edu.cn)

Many natural and artificial materials are hybrids interfaced by complexed nanostructures which impart them with interesting characteristics and limitless diversities[1]. As an important example, surface ligands are an essential part of nanocrystal-ligands complexes and dramatically affect their properties, such as luminescence[2–4], electronic accessibility[5–8], stability[9], biological compatibility[10,11], and solution processibility needed for device fabrication[8,12]. For nanocrystal-ligands complexes with their inorganic core of several nanometers, the thermodynamics of dissolution are largely determined by the molecular structure and dynamics of their surface ligands[13]. Recently, conceptually new entropic ligands were introduced[8,13], which can boost the solubility of nanocrystal-ligand complexes and enables the large-scale printing of electronics and optoelectronics[14]. Importantly, entropic ligands are identified as a general solution for achieving excellent solubility of nanocrystals with diverse inorganic composition[14,15], which plays a decisive role in both synthesis and processing[16,17]. For instance, by simply mixing two common $n$-alkanoate ligands with distinguishable hydrocarbon-chain lengths, solubility of the resulting nanocrystal-ligands complexes could increase up to ~6 orders of magnitude in comparison with either of pure-ligand counterparts. Based on macroscopic measurements, entropic ligands are suggested to boost solubility of nanocrystal-ligands complexes by substantially reducing the dissolution enthalpy and allowing the Gibbs free energy to be dominated by the entropy[8,13].

Among two classes of entropic ligands, mixed ligands with distinguishable hydrocarbon-chain lengths are simple, versatile, and readily applicable to both synthesis and processing of nanocrystal-ligands complexes[14–17]. However, the molecular picture of mixed ligands is unknown, which makes design of an entropic ligand system purely empirical. Similar to other non-periodic nanostructures, the ligand monolayer on the surface of an inorganic nanocrystal remains an unsettled front line for today's characterization techniques. In particular, the surface ligands are disordered, dynamic, and with low electron-density contrast, which limit the accessibility of powerful diffraction, electron microscopy, and scanning probe microscopy techniques[18–20].

In this work, we devise an integration of advanced NMR methodologies to demonstrate the first comprehensive molecular image of entropic ligands consolidating both morphological and dynamical aspects. Our study predicts the extreme solubility of nanocrystal-ligands complexes with mixed ligands based on NMR-derived ligand partition and dynamics, and therefore establishes a direct connection between the molecular picture and macroscopic dissolution thermodynamics. Furthermore, our research demonstrates a versatile strategy based on NMR spectroscopy to explore and quantify nanostructures with intrinsic disorder and dynamics, such as therapeutic nanoparticles[21], polymer nanocomposites[22], nanoporous materials[23] and bone crystallites[24].

## Results

### Deciphering the partition of mixed ligands on nanocrystal surface.
CdSe nanocrystals with their sizes in quantum confinement regime (quantum dots) were adopted in this study, though entropic ligands are known to work on different types of inorganic nanocrystals[8]. Typically, monodispersed CdSe nanocrystals were synthesized with one type of $n$-alkanoate ligands, e.g., myristate with 14 carbon units (Fig. 1, top panel). CdSe nanocrystals with mixed ligands (i.e., entropic ligands) were obtained by ligand exchange of the CdSe-myristate complexes with a certain ratio of $n$-hexanoic acid in solution (Fig. 1, bottom panel). The size and the size distribution of CdSe nanocrystals (3.0 ±

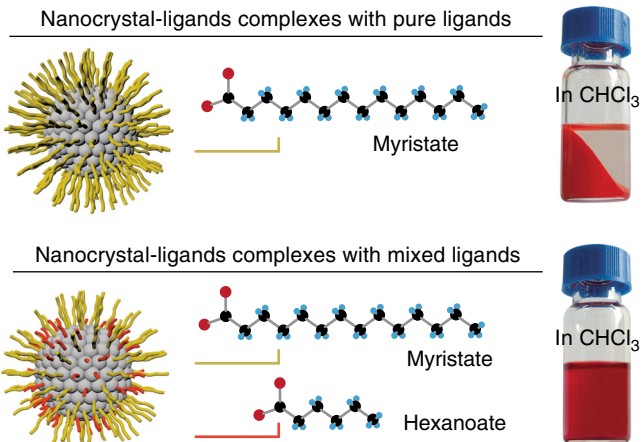

**Fig. 1** Solubility enhancement by mixing long and short ligands. The graphs illustrate nanocrystal-ligands complexes with pure ligands of myristate, or with mixed ligands of myristate and hexanoate. On the right: optical pictures showing the macroscopic differences between nanocrystal-ligands complexes with pure ligands and nanocrystal-ligands complexes with mixed ligands in chloroform solvent. The vials contain equal weight (0.25 g) of purified samples in 0.8 mL of chloroform. The nanocrystal-ligands complexes with mixed ligands dispersed completely while the nanocrystal-ligands complexes with pure ligands precipitate. The size of nanocrystal core is 3.0 ± 0.2 nm as determined by TEM

0.2 nm in diameter for the inorganic cores) remained the same before and after ligand exchange according to transmission electron microscopy (TEM) and ultraviolet–visible (UV–Vis) absorption measurements[25] (Supplementary Fig. 1). Prior to NMR investigations, all nanocrystal-ligands complexes were purified following existing procedures[8,13]. As shown in Fig. 1 (bottom panel), 0.25 g nanocrystal-ligands complexes with mixed ligands can be totally dispersed in 0.8 ml chloroform, yet the nanocrystal-ligands complexes with pure ligands are hardly dissolvable at room temperature.

In order to decipher the surface partition of different ligands, spin-labeled myristic acid with 100% $^{13}C$-enrichment on the carboxylate group was used. On a CdSe nanocrystal, the neighboring ligands were separated by ~0.5 nm in average at the anchoring point[26], which is under the radar of $^{13}C$-$^{13}C$ homonuclear dipolar coupling[27]. Normally, $^{13}C$-$^{13}C$ distances within 1 nm should be measurable[28]. When the nanocrystals coated with 100% $^{13}C$-labeled myristate ligands were partially substituted by unlabeled hexanoate (1.1% $^{13}C$ in natural abundance), the $^{13}C$-$^{13}C$ coupling network would be interrupted in a specific manner associated with the partition scheme.

Here we employed a solid-state NMR sequence named as center-band only detection of exchange (CODEX)[29], in which the relative intensity ($S/S_0$) versus NMR mixing time provides a sensitive and quantitative measure of the spin state including inter-spin coupling and dynamic behavior (Supplementary Fig. 2a–c). During the mixing time, $^{13}C$ spin diffusion and/or molecular reorientation change the chemical shift frequency and lead to the decay of signal intensity. We confirmed that the isolated carboxylate groups do not alter CODEX decay due to the rigidity of themselves (Supplementary Fig. 2c). Therefore, the measured CODEX decays were purely modulated by the $^{13}C$-$^{13}C$ coupling arising from the neighboring $^{13}C$-labelled ligands. We carried out numerical modeling by placing the ligands on a spherical surface uniformly or in bundles of different sizes (see Supplementary Fig. 2e, f for more details). The CODEX strategy is a viable pathway to distinguish distribution of ligands with similar molecular structures, and it could be a general

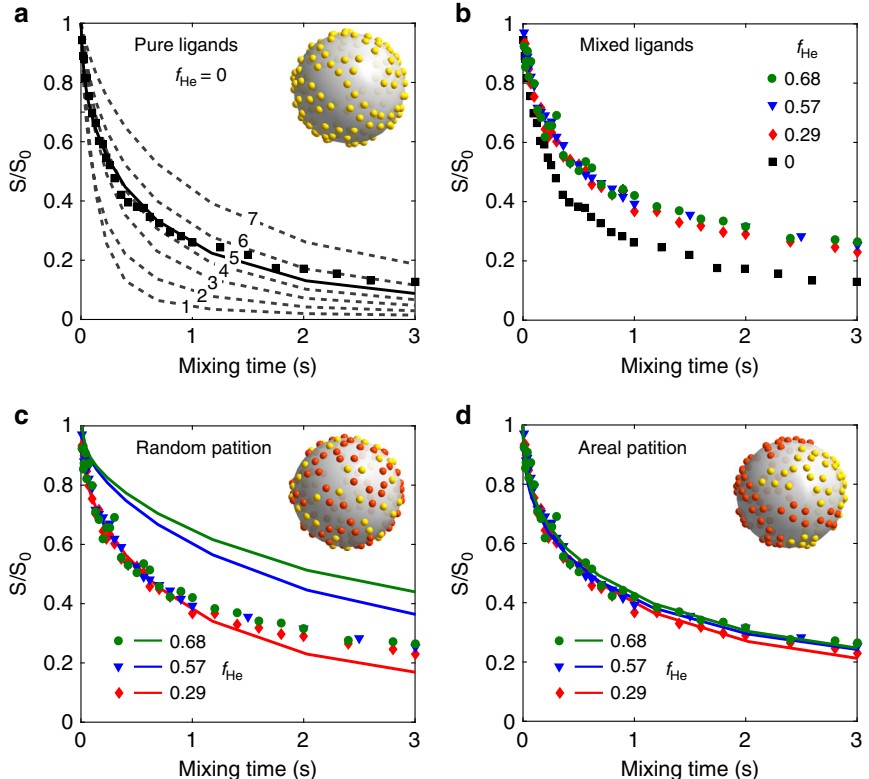

**Fig. 2** $^{13}$C CODEX results and ligand partition. **a** The CODEX decay of nanocrystal-ligands complexes with pure ligands based on the signal of $^{13}$C-labeled myristate ligands. The solid and dotted lines are the simulation results of ligand distributions with different numbers (1–7) of ligands in one bundle. The solid line corresponds to the scenario of five ligands in one bundle, which has the smallest deviation from the experimental data. **b** $^{13}$C CODEX decay of complexes with different hexanoate fractions. In **c** and **d**, the colored lines are simulations of CODEX decay curves of complexes with mixed ligands in (**c**) random partition and (**d**) areal partition. The yellow dots on the spherical nanocrystal core schematically represent the carboxylate groups of myristate and the red ones stand for the carboxylate groups of hexanoate. The spherical models used in CODEX simulation are 3 nm inorganic core with 135 ligands (the surface ligand density was determined by gas chromatography in Supplementary Table 1)

methodology for probing nanoscale atomic distribution. Compared to other methods[30–32], the spin-labeling is chemically non-disruptive and the sensitive dipolar interaction is much more amendable for quantitative modeling.

Figure 2a shows the CODEX decay of the nanocrystal-ligands complexes with pure myristate ligands (with $^{13}$C-labeled myristates), which provides the basis for our analysis. The double exponential feature can be described by a non-uniform distribution of small bundles consisting of ~4–6 ligands considering the structural heterogeneity of nanoparticles (See Supplementary methods and Supplementary Fig. 2e, f). This bundling effect could originate from the small facets or surface reconstructions on spheroidal nanocrystals[33]. The revelation of ligand bundles corroborates the earlier report which described the islands of ligands on nanocrystal surface[34], while the CODEX approach offers a more quantitative interpretation on bundle size.

After the ligand exchange, the CODEX decays (Fig. 2b) lifted evidently for various hexanoate fractions ($f_{He}$), indicating notable changes in the neighboring environment of residual $^{13}$C-labeled myristates. Interestingly, while significantly different from that with pure myristate ligands, all CODEX decay curves for the nanocrystal-ligands complexes with mixed ligands in various fractions of hexanoates are quantitatively similar to each other (Fig. 2b).

To model our mixed-ligand systems, we invoked two general ligand partition schemes with fundamentally distinct features, namely ligands distributed in random (Random, Fig. 2c) and areal segregation of either myristates or hexanoates (Areal, Fig. 2d). Matching of experimental CODEX curves for different hexanoate fractions clearly identified the areal partition of surface

ligands. Therefore, our NMR investigation successfully realized a geometric classification of surface morphology of nanocrystal-ligands complexes with mixed ligands.

**Revealing the dynamic picture of surface ligands**. Though the partition of ligands may have profound implications on the ligand–ligand interactions, it is still a steady (or average) picture of the ligand monolayer around each inorganic nanocrystal core. Given the dominating role of entropy, steady picture is insufficient for understanding the molecular nature on solubility of nanocrystal-ligands complexes. However, as shown in the case of protein, the partition of ligands leads to a notable effect on their dynamic behaviors[35].

Deuterium ($^2$H) NMR quadrupolar pattern is a versatile probe to identify different modes of segmental reorientation of organic molecules[27,36]. Figure 3a shows the respective $^2$H patterns (under 2 kHz MAS) of an individual $CD_2$ site which undergoes different dynamic modes probable for surface ligands, namely static, trans-gauche$^+$-gauche$^-$ (tgg) rotation, and cone diffusion (described in Supplementary methods, Supplementary Fig. 4). As we can see, each of the patterns are highly distinctive by their width and shape, e.g., the positions of their edges and horns. In general, the more flexible the sites are, the narrower the patterns would be.

We carried out $^2$H NMR measurements under variable temperatures on nanocrystal-ligands complexes with fully deuterated myristates and protonated hexanoates. Figure 3b shows two sets of representative $^2$H patterns obtained at 245 and 300 K (other temperatures in Supplementary Fig. 3a). Evidently,

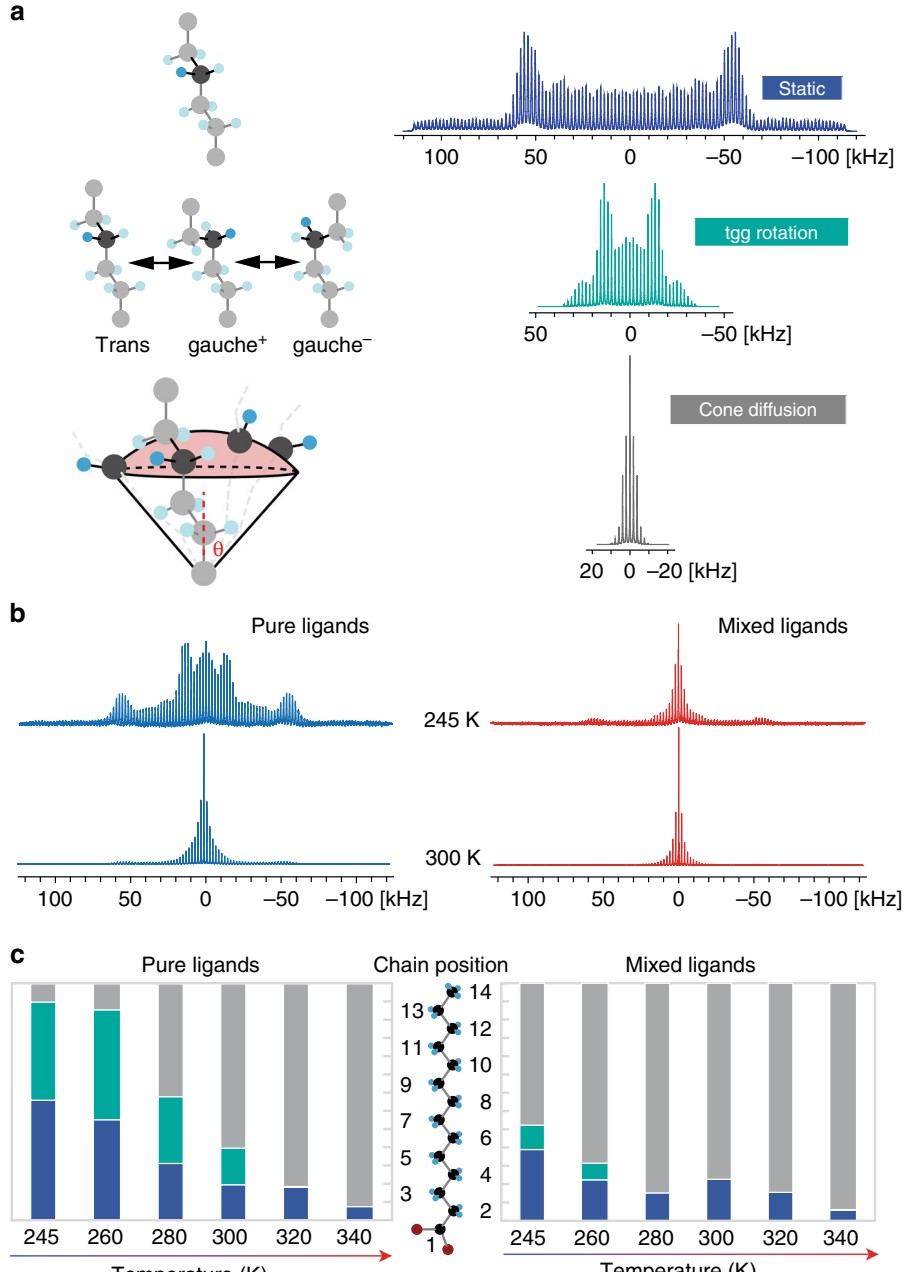

**Fig. 3** $^2$H NMR lineshapes and chain flexibility. **a** The three distinct dynamic modes of methylene units and the corresponding $^2$H NMR patterns under 2 kHz magic-angle spinning. These dynamic modes could present in a hydrocarbon chain at different temperatures or at different positions, e.g., the middle segment or the free end. **b** $^2$H NMR patterns for nanocrystal-ligands complexes with pure ligands ($f_{He} = 0$) and nanocrystal-ligands complexes with mixed ligands ($f_{He} = 0.68$) with fully deuterated myristates at 245 and 300 K. **c** The histograms of methylene flexibility along the myristate ligand at variable temperatures, based on the deconvolutions of $^2$H patterns. The blue, green and gray bars represent static deuterium, tgg rotation and cone diffusion, respectively

the differences in chain flexibility between different types of nanocrystal-ligands complexes are more pronounced at the lower temperature.

We deconvoluted each of the patterns in Supplementary Fig. 4 into the three dynamic modes described above with relative populations corresponding to the number of methylene units. Accordingly, we obtained the histograms of flexibility along the myristate chain at different temperatures (Fig. 3c). The myristates on the nanocrystal-ligands complexes with mixed ligands were found to be much more flexible at temperatures below 300 K, which is a clear indication of the weakening of ligand interactions and activation of chain dynamics. For nanocrystal-ligands

complexes with mixed ligands, most part of myristates switch into the cone diffusion mode at around 280 K, while for the nanocrystal-ligands complexes with pure ligands such transition only happens at around 320 K. The NMR-observed transition temperatures are consistent with the predicted melting points of the nanocrystal-ligands complexes (Table 1) and could explain the strong temperature dependence of solubility.

As a step further, we pursued site-specific quantification of ligand dynamics with the DIPSHIFT method[37], a two-dimensional NMR sequence resolving the $^1$H–$^{13}$C heteronuclear dipolar coupling of each carbon resonance. The averaging effect of $^1$H–$^{13}$C coupling is a quantitative reference for the segmental

**Table 1 The thermodynamic parameters of nanocrystal-ligands complexes obtained via light scattering (top panel) and via NMR-based calculations (bottom panel)**

| Hexanoate fraction | 0 | 0.05 | 0.68 |
|---|---|---|---|
| Dissolution enthalpy $\Delta^m H_{NC}$ (kJ mol$^{-1}$) | 308 | 304 | 265 |
| Dissolution entropy $\Delta^m S_{NC}$ (J mol$^{-1}$ K$^{-1}$) | 874 | 892 | 916 |
| Melting point, $T_m = \Delta^m H_{NC}/\Delta^m S_{NC}$ (K) | 354 | 343 | 291 |
| Total interaction energy $E_{tot}$ (kJ mol$^{-1}$) | 305 | 304 | 278 |
| Ligand-ligand interaction energy $E_{ligand}$ (kJ mol$^{-1}$) | 303 | 302 | 274 |
| Inter-particle interaction energy $E_{core}$ (kJ mol$^{-1}$) | 1.5 | 1.5 | 3.9 |

motion of surface ligands. The DIPSHIFT sequence takes advantage of favorable $^{13}$C spectral resolution and does not require isotope enrichment. Figure 4a shows the theoretical DIPSHIFT curves for different $^1$H–$^{13}$C coupling strengths in the fast motion regime. The observed coupling strength, i.e., the residual dipolar coupling, can be converted into the opening angles of cone diffusion model (Supplementary Fig. 5c). A shallower dip corresponds to a weaker coupling, and therefore a larger opening angle[38].

Figure 4b presents the $^{13}$C spectra of ligands of nanocrystal-ligands complexes with either pure myristate or mixed ligands, where resolvable signals are assigned to the segments of

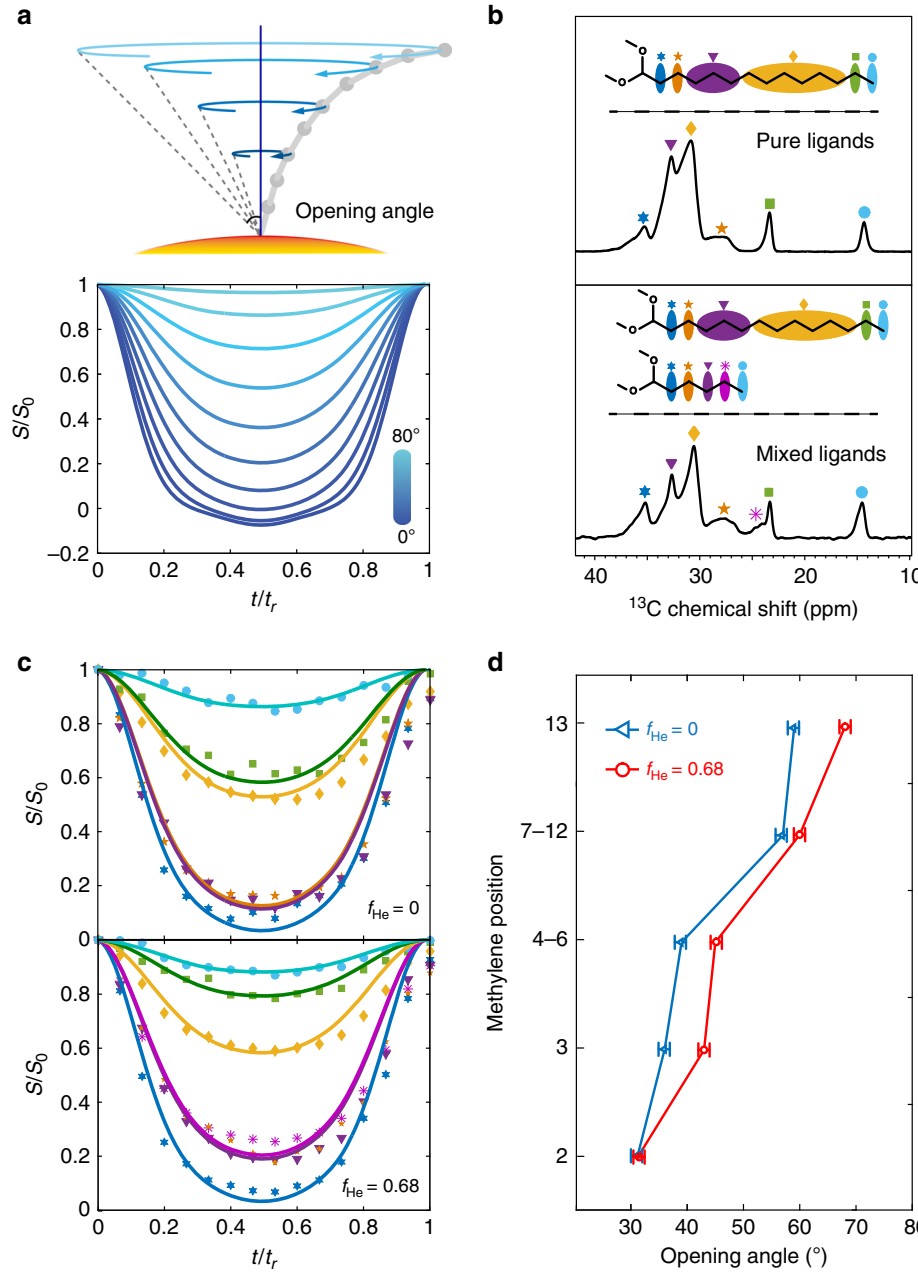

**Fig. 4** $^1$H-$^{13}$C DIPSHIFT results and opening angles. **a** The cone diffusion model of methylenes and the simulated DIPSHIFT curves for different opening angles. **b** $^{13}$C solid-state NMR spectra obtain at 300 K. The peaks correspond to the chain segments labelled by the same symbols. **c** The DIPSHIFT results (dots) and fitting curves (solid lines). The colors correspond to the chain segments labelled in (**b**). **d** The opening angles of each chain segment of myristate ligands assuming the cone diffusion motion. The fitting standard deviation is smaller than one degree. The methylene positions are defined in Fig. 3c

*n*-alkanoates (inset, Fig. 4b). From head to tail along the hydrocarbon chain, the depth of the dip decreases gradually, confirming an increasing flexibility towards the end of methyl group (Fig. 4c). After theoretical conversion of the measured $^1$H–$^{13}$C coupling, the opening angles of each myristate segment were obtained for different ligand fractions. Figure 4d shows that the opening angles of myristates on nanocrystal-ligands complexes with mixed ligands ($f_{He} = 0.68$) were found to be much wider than those on nanocrystal-ligands complexes with pure ligands ($f_{He} = 0$), confirming substantially weakened ligand–ligand interaction and significantly enhanced chain dynamics.

### Predicting the solubility based on ligand–ligand interactions.

Our earlier work based on macroscopic measurements[13] revealed that the dissolution of nanocrystal-ligands complexes is equivalent to a two-step process. In the first step, the solid is melted, which is accompanied by dramatic changes in the enthalpy and intramolecular entropy. In the second step, the melted solid and solvent, two liquids, are mixed, which involves mostly the ideal entropy change of mixing. As long as the inorganic core is relatively small (<~5 nm) and the hydrocarbon chain is reasonably long, the enthalpy of dissolution ($\Delta^m H_{NC}$) of the entire process is dominated by the destruction of ligand–ligand interactions. At the same time, a large amount of intramolecular conformational entropy ($\Delta^m S_{NC}$) would be released (Table 1).

Based on the molecular pictures revealed by our NMR studies, we would like to show that the macroscopic solubility of nanocrystal-ligands complexes can be predicted directly from their molecular partition and dynamics. We first predicted the interaction energy ($E_{ligand}$) using the dispersion energy model for hydrocarbon chains[39]. Such calculations were grounded on the fact that the free volume of each methylene unit (Fig. 5a) can be quantified by the opening angle determined by DIPSHIFT experiments. In addition, the calculation considered the interparticle interaction of nanocrystal cores ($E_{core}$)[40] although it makes up a relatively small contribution to the total interaction energy ($E_{tot}$). The results (described in Supplementary methods, Supplementary Fig. 6) showed that the total interaction energy is largely equivalent to dissolution enthalpy (Table 1). Our calculation ultimately predicted the solubility values for a range of nanocrystal-ligands complexes with mixed ligands at the room temperature, which agree well with the measured values (Fig. 5b).

Our work reached a revealing conclusion that the exceptional solubility of nanocrystal-ligands complexes with entropic ligands is quantitatively dictated by the dynamic behavior of ligands along with their partition on the surface of a nanocrystal. The molecular picture established in this work serves as a theoretical blueprint for the flourish of entropic ligands in the field of colloidal nanocrystals. Moreover, our NMR methodology will be applicable to diverse disordered and dynamic nanostructures and could provide crucial guidance for the dedicated regulation of their surface properties.

## Methods

**Synthesis of CdSe-ligands complexes.** The synthesis of CdSe nanocrystals was performed by injecting a 1.0 mL Se-octadecene suspension (0.2 mol L$^{-1}$) into a hot (250 °C) mixture of CdO, myristic acid and 1-octadecene in a 50 mL three-neck flask. Needle-tip aliquots were taken for UV–vis and photoluminescence measurements to monitor the reaction until the desired size has been reached[41]. The reaction mixture of the CdSe-ligands complexes was further purified according to the procedures described in Supplementary methods, and infrared measurement verified that remaining ODE and free acids had been fully removed.

**Preparation of nanocrystal-ligands complexes with mixed ligands.** Ten milligram purified complexes with pure myristate ligands were dissolved in 0.5 mL chloroform in a 4 mL vial and kept at 50 °C as a clear solution. Hexanoic acid with molar ratios ranging from 0.1 to 2 relative to bonded myristate ligands was added into solution for 2 h. The resulting nanocrystal-ligands complexes with mixed ligands were purified, and the solids have been vacuumed for 12 h to remove residual solvents.

**Measurement of ligand fractions.** The fraction of hexanoate on nanocrystal-ligands complexes with mixed ligands (Supplementary Table 1) was determined by gas chromatography. The measurements were carried out on samples digested by saturated hydrochloric acid. The molar ratio of hexanoic acid to myristic acid in the digested solution is assumed to be the same as the ratio of those two ligands on nanocrystal surface.

**Surface density of ligands.** The hydrogen and carbon mass fractions of nanocrystal-ligands complexes were determined by elemental analysis of purified samples. The surface density of nanocrystal-ligands complexes with pure ligands was determined to be ~135 ligands per crystal according to the formula provided in the Supplementary methods. The ligand densities for nanocrystal-ligands complexes with mixed ligands are about the same as their pure-ligand precursors as shown in Supplementary Table 1.

**Solubility measurement.** The solubility at room temperature was determined by the UV–vis absorbance of saturated solutions of CdSe complexes. The solubility of complexes at various temperatures was measured by the scattering method[26], in which a known concentration of dissolved complexes in chloroform was slowly cooled down from a relatively high temperature. The scattering intensity of 750 nm

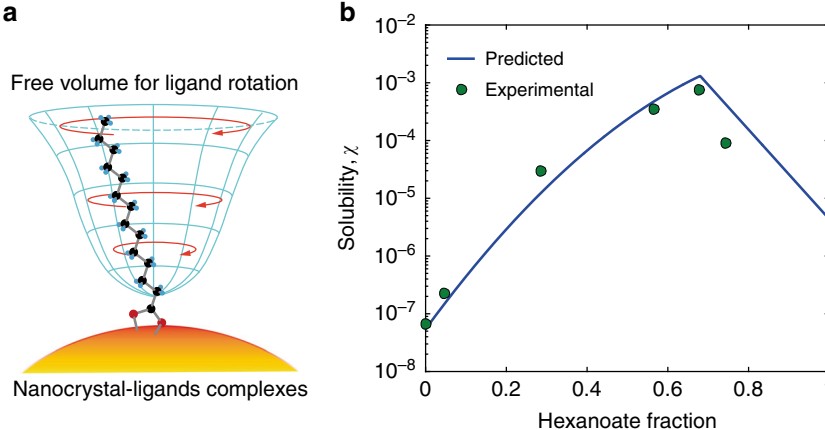

**Fig. 5** Predicting ligand interaction energy and solubility. **a** Graphical illustration of the geometric confinement of a flexible surface ligand in the dispersive energy force field. **b** The predicted solubility at room temperature (blue solid line) and the experimental values measured in chloroform based on ultraviolet absorbance (green circles)

laser shows a sudden jump when the concentration reaches the solubility at the specific temperature.

**NMR experiments**. $^{13}$C CODEX experiments were carried out on a Bruker Avance III HD 600 MHz spectrometer using a 1.3 mm triple channel magic-angle spinning (MAS) probe. The spinning speed was 8 kHz and $^{13}$C chemical shift was referenced to the adamantine signal at 38.5 ppm on the tetramethylsilane (TMS) scale. The $^{2}$H experiments were performed on a Bruker Avance III HD 600 MHz spectrometer using the solid echo pulse sequence under 2 kHz MAS or under static conditions. The $^{13}$C–$^{1}$H DIPSHIFT experiments were carried out on a Bruker Avance III HD 400 MHz spectrometer using a 3.2 mm triple channel MAS probe with a spinning speed of 4431 Hz (calculated from the $^{1}$H homonuclear decoupling strength). Typical radio frequency field strengths were 62.5 kHz for $^{13}$C, 100 kHz for $^{2}$H and 100–115 kHz for $^{1}$H. The magic angle and field homogeneity of the spectrometers were optimized with KBr and adamantine, respectively. The temperature of NMR experiments was controlled by the Bruker BCU II unit.

**Modelling of ligand partition and interactions**. The detailed analysis of CODEX and DIPSHIFT experiments and the methods for numerical modelling are described in the Supplementary methods.

## Data availability
The data that support the findings of this study are available from the corresponding authors upon reasonable request.

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

## Acknowledgements
This work was supported by National Key Research and Development Program of China 2016YFA0203600 and 2016YFB0401600, the National Natural Science Foundation of China No. 21761142009, and Zhejiang Provincial Natural Science Foundation R19B050003.

## Author contributions
X.K. and X.P. planned the project and finalized the manuscript. Z.P. performed the SSNMR experiments, modelling and drafted the manuscript. J.Z. and W.C. prepared the materials and carried out characterizations.

## Additional information

**Competing interests:** The authors declare no competing interests.

