## [Peer Review File · Nature Communications]

Reviewers' comments:

Reviewer #1 (Remarks to the Author):

This manuscript reports applying a variety of state-of-the art solid-state NMR methods (variable temperature 2H NMR, CODEX and DIPSHIFT) to study the surface structure of nanomaterials and ligand-ligand interactions at atomic level. The authors show areal partition of surface ligands on CdSe nanocrystals, which explains the much larger solubility of the mixed-ligand nanocrystals. It is directed related to the applications of these materials, while few approach can give such information. I find the authors' presentation of using solid-state NMR spectroscopy to illustrate the related problems very interesting and valuable. There are a few major and minor points to address before the article can be accepted for publication in Nature Communications.

Major issues:

Page 5, line 101, the deconvolution of different patterns should be provided (currently only one example is shown, Supplementary Figure 3d)

The authors generated ligand partition models (Supplementary Figure 2d). However, the readers wonder how representative are. In case the ligands are not in a configuration as the authors propose, are the results going to be different. For example, 4 ligands in one bundle, if ligands form "squares like bundle" instead of "diamonds like bundle", will the simulation be different?

Minor issues:

Page 4, line 73, CODEX is "the only pathway to distinguish distribution of ligands with highly resemble molecular structures". I doubt it is the ONLY one. Is there any reference?

Page 4, line 77, Figure 1a shows small bundles consisting of "approximately 5 ligands". I don't think Figure 1a is very clear. The dots are overlapping with 4 and 5 in a mixing time of less than 0.5 s, while it is more like with 6 ligands at a high mixing time of 2-3 s. Does the fitting with shorter or longer mixing time give a better estimation of the numbers in the bundle? I think shorter and longer mixing time seem to provide different estimates.

Page 4, line 87-89, "The partition of ligands may have profound implications on the mechanisms of ligand exchange processes and consequently lead to a notable effect on their dynamic behaviors." Are there any references for this?

Figure 1, different colors are used for ligands shown in Figure 1c and 1d. It is better to show what yellow and orange color stand for.

Figure 2, the authors should use larger figures. This also applies to Figure 3 and 4, Supplementary Figure 1a. It is not very clear with small figures.

Figure 3, why the open angles for the methylene positions=2 and 14 are the same for pure and mixed ligand samples? It is hard to understand. 2 and 14 are very different in nature. 2 is very close to the surface of nanocrystal and should be most rigid while 14 is on the outside and should be most mobile. Also, the figure caption of Figure 3d should mention the methylene position is defined in Figure 2.

Another questions is the simulation in Figure 3a always show symmetric curves and S/S0 is always 1 with $t/t_r=1$. However, the experimental data (Figure 3c and Supplementary Figure 4b) seem to always give a smaller S/S0 than expected at a high t/t_r . Why?

Figure 4, Are Figure 4c optic pictures, or just schematic representations (vials and tubes)? They don't look like so because there is no scale and are too small. But the figure caption states that there are 0.25 g complexes so they should be optic pictures. Also it's better that the authors use the same colors (yellow and orange) for the two ligands as in Figure 1?

The figure caption do not match Supplementary Figure 3d and 3e. In particular, there is no supplementary figure 3e at all. This should be fixed.

Reviewer #2 (Remarks to the Author):

Pang et al. report on a new strategy to probe ligand partitioning on the surface of semiconductor (CdSe) nanoparticles using CODEX NMR. The authors are motivated to understand how using "entropic" mixed ligand shells and subsequent ligand arrangement on the particle surface impacts solubility of CdSe nanocrystals. The results are of broad interest to the nanoparticle and materials science communities and will dramatically affect how these materials are designed and used in catalysis, medicine, and energy systems. The NMR-based study allows the authors to understand both how the ligands are arranged on the nanoparticle surface as well as the resulting ligand mobility. However, as the paper is written now, the motivation of the work is not clear to the broad readership of Nature Communications. While the NMR measurements are carefully done and thoroughly explained, the reviewer has major concerns regarding how the models were constructed to simulate the CODEX results. As it stands now, the results cannot be reproduced nor can the reader ascertain whether or not the current results were properly interpreted.

Major Concerns:

Comment 1:

If the reader is not familiar with the authors' previous work on "entropic" mixed ligand shells, the motivation for understanding the relationship between ligand arrangement and dynamics on the surface of CdSe nanocrystals is lost until the very end of the paper. I suggest the authors move Figs 4b and c to the forefront of the manuscript to motivate the NMR study, which is a beautiful approach to understanding this phenomenon.

Comment 2: Page 3, line 62

The authors state that CdSe nanocrystals are typically 3 nm in diameter. Can the authors please add error bars to this measurement and the number of particles measured? This is key to understanding particle dispersity and interpreting the NMR results. Does the size change after ligand exchange with the hexanoate?

Comment 3: Page 4, line 78

The authors suggest that the ligands bundle on the surface of spheroidal nanocrystals due to small surface faceting. I'm having trouble visualizing this. Can the authors provide a reference that suggests this happening (perhaps a computational model)? I would expect that the higher energy vertex/edge sites with more dangling bonds to be passivated prior to the terrace sites on the surface.

Comment 4: Ligand partition model

The details of the ligand partition model used to simulate the CODEX results were unclear, and as result, it is difficult to determine if the authors' interpretation is reasonable. In the SI, the authors state that models were generated with an "iteration method." This language is confusing and suggests that first principle modeling was done when it was not.

As it stands now, the model cannot be reproduced. It is not clear what size sphere was used nor what the relative size of the ligand was (the more important parameter here). How was the size that the ligand occupies on the surface of the sphere determined? The authors present careful characterization of ligand quantification and ligand densities using GC-MS that could have been fed into the model, but there is no mention of it.

It is reasonable to not re-optimize the models after bundling. However, what was the rationale for choosing the inter-ligand distance of 0.36 nm? How did the authors define the "center point"?

Comment 5: Page 5, line 100

The authors state "the differences in chain flexibility are more pronounced at the lower temperature." Instead, I suggest the authors interpret this in terms of energy barrier to methylene rotation (which is what they are probing with the variable temperature measurement)?

Comment 6: Page 6, line 117

Can the authors please define "opening angle" or include an explicit picture of the angle and a description in the caption of Figure 3? In Fig 3a, it is not clear how the angle they have drawn is related to the surface of the particle and the ligand. Without this information, it is not clear if their interpretation is correct. On particular surface facets, a certain degree of tilting is expected for tight packing. For example, thiols are tilted ~30 degrees from normal on Au(111). What is expected for CdSe and how is this related to the observations from NMR?

Comment 7: Figure 3

The DIPSHIFT curves appear to show different trends for myristate and hexanoate-containing ligand shells. Yet, the open angles are similar in Fig 3d. I recommend the authors include error bars based on the DIPSHIFT fits from 3c.

Minor Concerns:

Comment 1:

Throughout the manuscript the authors use the terms "nanocrystal" and "complex" interchangeably. This terminology is incredibly confusing. Please replace every instance of "complex" with "nanocrystal" or "nanoparticle". When the term complex is used it implies an inorganic complex (e.g. nanoparticle precursor, standard used for comparison, etc).

Comment 2:

Early NMR work from Pines and Alivisatos (10.1016/0009-2614(92)80023-5) suggested ligand bundling on the surface of CdS quantum dots. The authors should cite this and other relevant work. Work using NMR (¹³C and ²H NMR) by Reven and coworkers to study dynamics seems relevant to the ²H studies here. The authors have the opportunity to discuss how their work provides a fresh approach to understanding ligand arrangement on the surface of nanocrystals in a quantitative fashion that was not leveraged previously.

Comment 3:

I suggest that the authors add in the molecular structure of both ligands (myristic acid and hexanoate) to the first figure so the reader can readily ascertain the inter-ligand interactions that they would expect for mixed ligand systems.

Comment 4: Page 3, Line 64

The language "under the radar of ¹³C-¹³C homonuclear dipolar coupling" is confusing. From this statement, it is not clear if ligands can or cannot be detected via dipolar coupling. Presumably, the authors mean that 0.5 nm is close enough that the ligands should exhibit dipolar coupling with one

another.

Comment 5: Page 4, Line 70

The authors state that the relative intensity vs mixing time provides a sensitive and quantitative measure of the "spin state." The language of "spin state" is far too vague for the reader to understand what the CODEX experiment is evaluating. The authors do a nice job of explaining CODEX in the SI – I would suggest incorporating some of this into the main text for clarity.

Comment 6: Page 4, Line 71

The phrase "highly resemble molecular structures" does not make sense. Perhaps the authors mean "similar molecular structures."

Comment 7: Figure 1 caption

The authors need to clarify the difference between the solid and dotted lines in Fig 1a (the reader has to assume that the solid line is also a simulation, but just a best fit of the experimental data). It would also be helpful if the authors listed the size (with error bars) of the nanocrystals for each panel.

Comment 8: Page 5

The authors need a transition for going from looking at ligand arrangement with CODEX to examining ligand dynamics. The motivation to understanding the dynamics is not clear in the way it is written right now.

Comment 9: Figure 2 caption

What are the three species whose 2H spectra shown in Fig 2a showing the three distinct modes of methylene units?

Reviewer #3 (Remarks to the Author):

Pang, Zang et al. demonstrate (1) how isotopic enrichment of the ligand of the NC in combination with the use of CODEX type experiments lead to quantitate the distribution of ligands at the surface of NCs capped with a mixture of ligands. (2) They demonstrate how the use of deuterium NMR at variable temperatures permits to investigate the dynamics of the ligands at the surface of NCs such as the flexibility of the ligand, i.e. myristate chain used here. (3) They show that by using DIPSHIFT method they can also quantify site specific dynamics of the ligands. The dynamics of the ligands is compared for pure-and mixed-ligand particles. Notably, a higher flexibility of the chain is observed for mixed-ligand complexes. Finally, these achievements of determining the partition of ligands at the surface of NCs and their dynamics can be directly related to the macroscopic solubility of NCs. The results are used to explain the increased solubility of mixed-ligand complexes.

The manuscript is well-written and the NMR data are supporting the conclusions drawn by the authors. In particular, I appreciated the use of the CODEX experiments to deduce the partition of the different ligands the surface of the NC.

In my opinion the main weakness of the manuscript is that the experiments are demonstrated on model ligand-NC complexes and using specific ¹³C and ²H isotope labelling, while in the introduction the authors advertise the versatility of their approach and the opening towards interesting applications (line 51-52 page 3.) My opinion is that the proposed strategy to investigate ligand dynamics relies on difficult experiments to implement if the ligands are diluted in the NC or if they are not model compounds... What are the experimental time of these experiments CODEX and DIPSHIFT?

The 2H studies as well as the DIPSHIFT experiments clearly highlight differential dynamics between the pure- and mixed-ligand NCs. The authors explain this observation by weakened ligand-ligand interactions in the case of the mixed-ligand complexes. I would suggest to attenuate this claim. For example weaker interactions with the surface of the NCs could also be considered.

Response to the reviewers

Reviewer #1 (Remarks to the Author):

This manuscript reports applying a variety of state-of-the-art solid-state NMR methods (variable temperature ^2H NMR, CODEX and DIPSHIFT) to study the surface structure of nanomaterials and ligand-ligand interactions at atomic level. The authors show areal partition of surface ligands on CdSe nanocrystals, which explains the much larger solubility of the mixed-ligand nanocrystals. It is directly related to the applications of these materials, while few approaches can give such information. I find the authors' presentation of using solid-state NMR spectroscopy to illustrate the related problems very interesting and valuable. There are a few major and minor points to address before the article can be accepted for publication in Nature Communications.

Major issues:

Page 5, line 101, the deconvolution of different patterns should be provided (currently only one example is shown, Supplementary Figure 3d)

Answer: As the reviewer suggested, we have added the deconvolution results of ^2H NMR patterns at all experimental temperatures to Supplementary Figure 4

The authors generated ligand partition models (Supplementary Figure 2d). However, the readers wonder how representative are. In case the ligands are not in a configuration as the authors propose, are the results going to be different. For example, 4 ligands in one bundle, if ligands form “squares like bundle” instead of “diamonds like bundle”, will the simulation be different?

Answer: We have now included both square-like and diamond-like bundles in Supplementary Figure 2d-2f. We found that the CODEX decay is more relevant to the average distances between ligands in a bundle but not the exact arrangement, which is in accordance with the principle. To match the CODEX decay, the average numbers of ligands in square-like bundles and diamond-like bundles are both in the range of 4 ~ 6.

Minor issues:

Page 4, line 73, CODEX is “the only pathway to distinguish distribution of ligands with highly resemble molecular structures”. I doubt it is the ONLY one. Is there any reference?

Answer: Thanks for the comment. We have replaced “*the only pathway to distinguish distribution of ligands with highly resemble molecular structures*” by “*The CODEX strategy is a viable pathway to distinguish distribution of ligands with similar molecular structures*” (Page 5 line 92-95, Marked manuscript))

Page 4, line 77, Figure 1a shows small bundles consisting of “approximately 5 ligands”. I don't think Figure 1a is very clear. The dots are overlapping with 4 and 5 in a mixing time of less than 0.5 s, while it is more like with 6 ligands at a high mixing time of 2-3 s. Does the fitting with

shorter or longer mixing time give a better estimation of the numbers in the bundle? I think shorter and longer mixing time seem to provide different estimates.

Answer: Although a bundle with 5 ligands is not a perfect fit to the experimental data, it gives the smallest standard deviation compared to the rest. At short or long mixing times, bundles of 4 or 6 also show a reasonable fit. In fact, structural heterogeneities between individual nanocrystals or within a single nanocrystal are inevitable. On average, a distribution of 4-6 ligands in a bundle should provide a more realistic picture of the nanocrystal-ligands complexes. We have updated the statement in the manuscript. (Page 5 line 100-105, Marked manuscript))

Page 4, line 87-89, “The partition of ligands may have profound implications on the mechanisms of ligand exchange processes and consequently lead to a notable effect on their dynamic behaviors.” Are there any references for this?

Answer: We have replaced the original sentence with a more thorough transition paragraph in Page 6 line 117-122, Marked manuscript, and the reference 35 has been added.

Figure 1, different colors are used for ligands shown in Figure 1c and 1d. It is better to show what yellow and orange color stand for.

Answer: The yellow dots on the sphere models stand for the carboxylate group of myristate and the red ones stand for the carboxylate group of hexanoate. We have added these descriptions in the caption of Figure 2.

Figure 2, the authors should use larger figures. This also applies to Figure 3 and 4, Supplementary Figure 1a. It is not very clear with small figures.

Answer: Thank you for the recommendations. These figures have been updated.

Figure 3, why the open angles for the methylene positions=2 and 14 are the same for pure and mixed ligand samples? It is hard to understand. 2 and 14 are very different in nature. 2 is very close to the surface of nanocrystal and should be most rigid while 14 is on the outside and should be most mobile.

Answer: Methylene position No. 2 is the closest one to the nanocrystal surface and its motion has been locked due to the immobility of nanocrystals and possibly the surface interactions. The situations are the same for both long and short ligands, since they are all carboxylates bonded to CdSe nanocrystals of the same size. Methyl position No. 14 is the free end of the myristate and its motion combines fast methyl rotation with large-amplitude cone diffusion. In DIPSHIFT, the methyl groups give almost the same C-H couplings in both pure- and mixed-ligand complexes. However, it might not be fully accurate to describe the motion of methyl group using the opening angles in the same way as the rest methylene groups. Therefore, we removed the points of methyl groups in Figure 4d. In fact, the methyl groups contribute little to the ligand-ligand interaction energy.

Also, the figure caption of Figure 3d should mention the methylene position is defined in Figure 2.

Answer: As suggested by the reviewer, the figure caption has been updated.

Another question is the simulation in Figure 3a always show symmetric curves and S/S0 is always 1 with $t/t_r=1$. However, the experimental data (Figure 3c and Supplementary Figure 4b) seem to always give a smaller S/S0 than expected at a high t/t_r . Why?

Answer: Theoretically, S/S0 should be equal to 1 at $t/t_r=1$. However, the homonuclear coupling between protons could not be totally eliminated in solids due to high order Hamiltonians and equipment restrictions such as RF inhomogeneity. [Prog. Nucl. Magn. Reason. Spectrosc. 97, 1-39 (2016) doi: 10.1016/j.pnmrs.2016.08.001]. This will lead to a S/S0 smaller than 1 at $t/t_r=1$. Experimentally, perfect symmetric curves are hardly observed.

Figure 4, Are Figure 4c optic pictures, or just schematic representations (vials and tubes)? They don't look like so because there is no scale and are too small. But the figure caption states that there are 0.25 g complexes so they should be optic pictures. Also, it's better that the authors use the same colors (yellow and orange) for the two ligands as in Figure 1?

Answer: We have changed Figure 4c to the new Figure 1 with corrected captions. The pictures of vials and NMR tubes are all optic pictures. As the Reviewer suggested, the ligands in Figure 1 have been filled with the same colors as Figure 2

The figure caption do not match Supplementary Figure 3d and 3e. In particular, there is no supplementary figure 3e at all. This should be fixed.

Answer: Thank the Reviewer for correcting the error. We have fixed the figure number.

Reviewer #2 (Remarks to the Author):

Pang et al. report on a new strategy to probe ligand partitioning on the surface of semiconductor (CdSe) nanoparticles using CODEX NMR. The authors are motivated to understand how using "entropic" mixed ligand shells and subsequent ligand arrangement on the particle surface impacts solubility of CdSe nanocrystals. The results are of broad interest to the nanoparticle and materials science communities and will dramatically affect how these materials are designed and used in catalysis, medicine, and energy systems. The NMR-based study allows the authors to understand both how the ligands are arranged on the nanoparticle surface as well as the resulting ligand mobility. However, as the paper is written now, the motivation of the work is not clear to the broad readership of Nature Communications. While the NMR measurements are carefully done and thoroughly explained, the reviewer has major concerns regarding how the models were constructed to simulate the CODEX results. As it stands now, the results cannot be reproduced nor can the reader ascertain whether or not the current results were properly interpreted.

Major Concerns:

Comment 1:

If the reader is not familiar with the authors' previous work on "entropic" mixed ligand shells, the motivation for understanding the relationship between ligand arrangement and dynamics on the surface of CdSe nanocrystals is lost until the very end of the paper. I suggest the authors move Figs 4b and c to the forefront of the manuscript to motivate the NMR study, which is a beautiful approach to understanding this phenomenon.

Answer: Excellent suggestion! We have made Figure 4c as the new Figure 1 showing the concept of entropic ligands using mixed ligands. In addition, we have added some more background information of "entropic ligands" (Page 2 Line 37-44, Page 3 Line 64-67, Marked manuscript).

Comment 2: Page 3, line 62

The authors state that CdSe nanocrystals are typically 3 nm in diameter. Can the authors please add error bars to this measurement and the number of particles measured? This is key to understanding particle dispersity and interpreting the NMR results. Does the size change after ligand exchange with the hexanoate?

Answer: In Supplementary Figure 1b, we have added a statistical histogram of the diameters of nanocrystals measured in a TEM image. A total of 425 nanoparticles have been measured and the size distribution is nearly monodisperse (± 0.2 nm). Alternatively, the diameter and size distribution of nanocrystals can also be determined by UV-Vis spectra (Supplementary Figure 1e) according to the relationship between lowest-energy absorption peak with the particle size [Chem. Mater. 15, 2854-2860 (2003) doi:10.1021/cm034081k]. As shown in Supplementary Figure 1e, the UV absorption spectra are with the same sharp features and the lowest-energy absorption peak is 552 ± 2 nm upon ligand exchange with various ligand ratio. These results indicate the particle size and its distribution remained the same after ligand exchange.

Comment 3: Page 4, line 78

The authors suggest that the ligands bundle on the surface of spheroidal nanocrystals due to small surface faceting. I'm having trouble visualizing this. Can the authors provide a reference that suggests this happening (perhaps a computational model)? I would expect that the higher energy vertex/edge sites with more dangling bonds to be passivated prior to the terrace sites on the surface.

Answer: In general, quantum dots are regarded as polyhedral particles truncated by multiple facets [Rogach, Andrey L. "Semiconductor nanocrystal quantum dots." Wien-New York: Springer (2008)]. The ligands stabilize and reconstruct the surface atoms which made the surface arrangement different from the core [J. Am. Chem. Soc., 132, 3344-3354 (2010), doi:10.1021/ja907511r]. Unfortunately, for the nanocrystals of 3 nm, they are too small to use the high-resolution TEM to resolve surface structure. Therefore, the surface of quantum dots is highly

complex and remains elusive at the current stage. Our present study does not directly probe the surface morphology and therefore cannot answer the exact reason which contributes to the ligand bundling.

We modified the original sentence to “*This bundling effect could originate from the small facets or surface reconstructions on spheroidal nanocrystals [J. Am. Chem. Soc., 132, 3344–3354 (2010), doi:10.1021/ja907511r]*” (Page 5 Line 102-103, Marked manuscript)

Comment 4: Ligand partition model

The details of the ligand partition model used to simulate the CODEX results were unclear, and as result, it is difficult to determine if the authors’ interpretation is reasonable. In the SI, the authors state that models were generated with an “iteration method.” This language is confusing and suggests that first principle modeling was done when it was not.

Answer: We have provided more descriptions of CODEX in the manuscript and modified the text in the Supplementary materials. We did not use first principle simulation. Instead, we used the numerical modeling as we clarified in the manuscript. In the SI, the description has been modified to “*The initial distribution of pure ligands was generated by a numerical iteration method.*” (Page 5 Line 90-92)

As it stands now, the model cannot be reproduced. It is not clear what size sphere was used nor what the relative size of the ligand was (the more important parameter here). How was the size that the ligand occupies on the surface of the sphere determined? The authors present careful characterization of ligand quantification and ligand densities using GC-MS that could have been fed into the model, but there is no mention of it.

Answer: Great suggestion! We have added the structural information (size, ligand footprint, number of ligands per particle) into the revised manuscript to confirm that our models are fully consistent with the structural characterization (Fig. 2 caption, Page 18 Line 358-362, Marked manuscript)). Specifically, the diameter of CdSe nanocrystals is 3 nm as measured by TEM and UV-Vis (see Supplementary Figure 1). The average number of ligands on a single nanocrystal is 135 as determined by GC and elemental analysis (see Supplementary Table 1). Based on these measurements, we calculated the ligand footprint, i.e. the surface area occupied by single ligand, is 0.21 nm².

It is reasonable to not re-optimize the models after bundling. However, what was the rationale for choosing the inter-ligand distance of 0.36 nm? How did the authors define the “center point”?

Answer: The inter-ligand distance in a single bundle is 0.39 nm (not 0.36 nm) which gives the best fit of our experimental data. It is roughly in line with densely packed chelating ligands on a {100} facet of CdSe crystal, in which the inter-ligand is 0.43 nm. As we mentioned above, the exact reason for bundling is unclear, given the very complex surface structure of CdSe quantum dots. We tend not to overinterpret this value at the current stage. The center point is the geometric center of a bundle. They are shown in Supplementary Figure 2f as the red crosses.

Comment 5: Page 5, line 100

The authors state “the differences in chain flexibility are more pronounced at the lower temperature.” Instead, I suggest the authors interpret this in terms of energy barrier to methylene rotation (which is what they are probing with the variable temperature measurement)?

Answer : Our current measurements by ^2H NMR lineshape or ^1H - ^{13}C DIPSHIFT only provide the order parameter of the ligand motion, but not the motional rate. To measure activation energy, one must obtain the motional rates under different temperatures. In principle, the motional rate could be determined by NMR relaxation or other methods which could be pursued in the future research. The activation energy for hydrocarbon chains has been studied for a number of systems. For instance, the activation energy for chain rotation in different polyethylenes has been found to be around 10 to 40 kJ/mol [J. Polym. Sci. 1957, Vol. XXVI, 171-186 (1957)], and the activation energy for confined polyoxyethylene is also ~ 15 kJ/mol [Macromol. Chem. Phys. 206, 998–1005 (2005)]. We expect the rotation of ligands on the nanocrystals to have a comparable activation energy with those values.

Overall, we would agree that energy barrier should be useful to interpret chain flexibility at different temperatures. However, our current measurements could not determine the energy barrier. Thus, we would prefer just stating the facts at the moment.

Comment 6: Page 6, line 117

Can the authors please define “opening angle” or include an explicit picture of the angle and a description in the caption of Figure 3? In Fig 3a, it is not clear how the angle they have drawn is related to the surface of the particle and the ligand. Without this information, it is not clear if their interpretation is correct. On particular surface facets, a certain degree of tilting is expected for tight packing. For example, thiols are tilted ~ 30 degrees from normal on Au(111). What is expected for CdSe and how is this related to the observations from NMR?

Answer: Our current NMR measurements do not determine the relative angle between the crystal and ligand. Although thiols might be tilted on Au(111) surface, the angles of carboxylate on CdSe are not known so far. Given the binary composition of CdSe, we would guess the situation is much complex than the thiol-gold system. In particular, the CdSe QDs studied here does not have large flat facets, so we cannot assume the ligands are tightly packed. In a computational paper [Science 344, 1380-1384 (2014) doi:10.1126/science.1252727], the hydrocarbon chain of carboxylate ligands on PbS is shown roughly perpendicular to the facets. But we think this kind of static picture should be replaced with a dynamic picture which better illustrates the ligands under real situations.

In our context, opening angle is the measurement of motional amplitude of individual methylene units. We have updated Fig. 4a to better represent the opening angles.

Comment 7: Figure 3

The DIPSHIFT curves appear to show different trends for myristate and hexanoate-containing ligand shells. Yet, the open angles are similar in Fig 3d. I recommend the authors include error bars based on the DIPSHIFT fits from 3c.

Answer: As the Reviewer suggested that we have added the error bars of opening angle in Fig 4d.

Minor Concerns:**Comment 1:**

Throughout the manuscript the authors use the terms “nanocrystal” and “complex” interchangeably. This terminology is incredibly confusing. Please replace every instance of “complex” with “nanocrystal” or “nanoparticle”. When the term complex is used it implies an inorganic complex (e.g. nanoparticle precursor, standard used for comparison, etc).

Answer: Specifically, “nanocrystal” in this work represents just the inorganic part of QDs. The term “nanocrystal-ligands complex” refers to the entire structure of both inorganic core and bonded ligands. Considering the focus of this work, we think it would be a good idea to distinguish these two terms. To make it clear, we have provided a definition for the two terms in the Abstract (Page 1, Line 11-12, Marked manuscript) and avoided using “complex” throughout the manuscript.

Comment 2:

Early NMR work from Pines and Alivisatos (10.1016/0009-2614(92)80023-5) suggested ligand bundling on the surface of CdS quantum dots. The authors should cite this and other relevant work. Work using NMR (13C and 2H NMR) by Reven and coworkers to study dynamics seems relevant to the 2H studies here. The authors have the opportunity to discuss how their work provides a fresh approach to understanding ligand arrangement on the surface of nanocrystals in a quantitative fashion that was not leveraged previously.

Answer: Thanks for the suggestion. As the Reviewer suggested, the work by Pines and Alivisatos has been cited (Ref 34 in the Marked manuscript) and discussed in the revised manuscript. The ligand bundling corroborates the work by Pines and Alivisatos, while our CODEX approach offered a more quantitative interpretation on the bundle size. Reven’s work on nanoparticles using 2H and 13C NMR have also been cited (Ref 36 in the Marked manuscript) and discussed accordingly.

Comment 3:

I suggest that the authors add in the molecular structure of both ligands (myristic acid and hexanoate) to the first figure so the reader can readily ascertain the inter-ligand interactions that they would expect for mixed ligand systems.

Answer: Great suggestion! The molecular structures of myristic acid and hexanoic acid are now included in the new Figure 1 in the Marked manuscript.

Comment 4: Page 3, Line 64

The language “under the radar of ^{13}C - ^{13}C homonuclear dipolar coupling” is confusing. From this statement, it is not clear if ligands can or cannot be detected via dipolar coupling. Presumably, the authors mean that 0.5 nm is close enough that the ligands should exhibit dipolar coupling with one another.

Answer: We have added a statement for clarity. “Normally, ^{13}C - ^{13}C distances within 1 nm should be measurable.” ^{13}C - ^{13}C homonuclear dipolar coupling of spins 0.5 nm apart can be easily measured using the CODEX method. (Page 4 Line 78-79, Marked manuscript)

Comment 5: Page 4, Line 70

The authors state that the relative intensity vs mixing time provides a sensitive and quantitative measure of the “spin state.” The language of “spin state” is far too vague for the reader to understand what the CODEX experiment is evaluating. The authors do a nice job of explaining CODEX in the SI – I would suggest incorporating some of this into the main text for clarity.

Answer: As the Reviewer suggested, we have provided more descriptions of the CODEX method in manuscript. (Page 4 Line 85-92, Marked manuscript)

Comment 6: Page 4, Line 71

The phrase “highly resemble molecular structures” does not make sense. Perhaps the authors mean “similar molecular structures.”

Answer: We have replaced “highly resemble molecular structures” by “similar molecular structures” (Page 5 Line 93-95, Marked manuscript)

Comment 7: Figure 1 caption

The authors need to clarify the difference between the solid and dotted lines in Fig 1a (the reader has to assume that the solid line is also a simulation, but just a best fit of the experimental data). It would also be helpful if the authors listed the size (with error bars) of the nanocrystals for each panel.

Answer: The original Figure 1 now becomes Figure 2. All the solid and dotted lines are simulation results, while the solid line represents the one with the smallest standard deviation. Our numerical modeling is based on nanocrystals with a diameter of 3 nm. We have included more descriptions in the caption of Figure 2.

Comment 8: Page 5

The authors need a transition for going from looking at ligand arrangement with CODEX to examining ligand dynamics. The motivation to understanding the dynamics is not clear in the way it is written right now.

Answer: Ligand arrangement affects ligand-ligand interactions, and the ligand-ligand interactions manifest as the changes in ligand dynamics. By studying ligand dynamics, we can reveal

ligand-ligand interactions. We added a transition/justification paragraph (Page 7 Line 117-122, Marked manuscript).

Comment 9: Figure 2 caption

What are the three species whose 2H spectra shown in Fig 2a showing the three distinct modes of methylene units?

Answer: They represent a CH₂ unit in a segment of hydrocarbon chain which could undergo three different dynamic modes in different situations. The description can be found in the Marked manuscript (Figure 3 caption). *“These dynamic modes could present in a hydrocarbon chain at different temperatures or at different positions e.g. the middle segment or the free end.”* (Page 19 Line 367-369, Marked manuscript)

Reviewer #3 (Remarks to the Author):

Pang, Zang et al. demonstrate (1) how isotopic enrichment of the ligand of the NC in combination with the use of CODEX type experiments lead to quantitate the distribution of ligands at the surface of NCs capped with a mixture of ligands. (2) They demonstrate how the use of deuterium NMR at variable temperatures permits to investigate the dynamics of the ligands at the surface of NCs such as the flexibility of the ligand, i.e. myristate chain used here. (3) They show that by using DIPSHIFT method they can also quantify site specific dynamics of the ligands. The dynamics of the ligands is compared for pure-and mixed-ligand particles. Notably, a higher flexibility of the chain is observed for mixed-ligand complexes. Finally, these achievements of determining the partition of ligands at the surface of NCs and their dynamics can be directly related to the macroscopic solubility of NCs. The results are used to explain the increased solubility of mixed-ligand complexes.

The manuscript is well-written and the NMR data are supporting the conclusions drawn by the authors. In particular, I appreciated the use of the CODEX experiments to deduce the partition of the different ligands the surface of the NC.

In my opinion the main weakness of the manuscript is that the experiments are demonstrated on model ligand-NC complexes and using specific ¹³C and ²H isotope labelling, while in the introduction the authors advertise the versatility of their approach and the opening towards interesting applications (line 51-52 page 3.) My opinion is that the proposed strategy to investigate ligand dynamics relies on difficult experiments to implement if the ligands are diluted in the NC or if they are not model compounds... What are the experimental time of these experiments CODEX and DIPSHIFT?

Answer: Deuterated ligands and ¹³C labelled ligand are commercially available and the cost is affordable. Yes, the labelling strategy is common for NMR experimentalists. For instance, isotope-labeled amino acids are constantly used in the NMR studies of proteins. At the same time, isotope labeling generally does not change the chemical properties of the compounds. Many biological studies use isotope labeling strategies. e.g. ¹⁴C, ¹³C, ¹⁵N ...

It should be noted that the DIPSHIFT method does not require ^{13}C or ^2H labeling at all and it provides significant information. It can be a general approach for most kinds of organic ligands without special treatment.

The experimental time for a full set of DIPSHIFT experiments on our 9.4 T magnet is about 20 hours for ~ 0.2 mmol samples containing nature abundance ^{13}C spins (1% of total carbon). The measurement time for CODEX is similar but it requires ^{13}C enrichment. The time consumption is common for typical SSNMR experiments. To do qualitative comparison between different samples, this time can be reduced substantially.

The ^2H studies as well as the DIPSHIFT experiments clearly highlight differential dynamics between the pure- and mixed-ligand NCs. The authors explain this observation by weakened ligand-ligand interactions in the case of the mixed-ligand complexes. I would suggest to attenuate this claim. For example, weaker interactions with the surface of the NCs could also be considered.

Answer: Our previous results (Refs 8,13 in the Marked manuscript) revealed that carboxylates bonded to the surface of inorganic cores steadily and nearly close packed on the surface. As a result, the inter-particle interaction enthalpy is mostly determined by the ligand-ligand interactions between adjacent nanoparticles, which is particularly true for nanocrystal-complexes with relatively small inorganic cores. Furthermore, close-packed nature of the ligands on the surface on each inorganic core excludes direct contact of the ligands on one inorganic core to the surface of another inorganic core.

REVIEWERS' COMMENTS:

Reviewer #1 (Remarks to the Author):

After carefully reading the authors' revised manuscript and their response to the reviewers, I have no further questions. The manuscript can now be recommended for publication in Nature Communications.

Reviewer #2 (Remarks to the Author):

The authors have addressed my concerns. This manuscript is now suitable for publication in Nature Communications.